# Interventions to reduce interpersonal stigma towards patients with a mental dysregulation for ambulance and emergency department healthcare professionals: review protocol for an integrative review

Geurt Van de Glind [iD] ,[1] Niek Galenkamp,[1] Nienke Bleijenberg,[1] Lisette Schoonhoven,[2,3] Floortje E Scheepers,[4] Julia Crilly,[5,6] Mark van Veen,[1] Wietske H W Ham[1]

For numbered affiliations see end of article.

**Correspondence to**
Dr Geurt Van de Glind;
geurt.vandeglind@hu.nl

## ABSTRACT

**Introduction** Worldwide, there is an increase in the extent and severity of mental illness. Exacerbation of somatic complaints in this group of people can result in recurring ambulance and emergency department care. The care of patients with a mental dysregulation (ie, experiencing a mental health problem and disproportionate feelings like fear, anger, sadness or confusion, possibly with associated behaviours) can be complex and challenging in the emergency care context, possibly evoking a wide variety of feelings, ranging from worry or pity to annoyance and frustration in emergency care staff members. This in return may lead to stigma towards patients with a mental dysregulation seeking emergency care. Interventions have been developed impacting attitude and behaviour and minimising stigma held by healthcare professionals. However, these interventions are not explicitly aimed at the emergency care context nor do these represent perspectives of healthcare professionals working within this context. Therefore, the aim of the proposed review is to gain insight into interventions targeting healthcare professionals, which minimise stigma including beliefs, attitudes and behaviour towards patients with a mental dysregulation within the emergency care context.

**Methods and analysis** The protocol for a systematic integrative review is presented, using the Preferred Reporting Items for Systematic Review and Meta-Analysis Protocols recommendations. A systematic search was performed on 13 July 2023. Study selection and data extraction will be performed by two independent reviewers. In each step, an expert with lived experience will comment on process and results. Software applications RefWorks-ProQuest, Rayyan and ATLAS. ti will be used to enhance the quality of the review and transparency of process and results.

**Ethics and dissemination** No ethical approval or safety considerations are required for this review. The proposed review will be submitted to a relevant international journal. Results will be presented at relevant medical scientific conferences.

## STRENGTHS AND LIMITATIONS OF THIS STUDY

⇒ To capture as many interventions as possible, in different stages of development, a broad search strategy including a range of considered terms, types of research (qualitative and quantitative) and in ambulance and emergency department setting with no limits to the publication date, is chosen.
⇒ The quality of included papers will be assessed by two independent reviewers, using an appropriate instrument, to inform readers on the evidence-based strength or weakness of included papers.
⇒ A person with lived experience is part of the review process and will coauthor the review.
⇒ Publications in English and Dutch only are included, and important publications in other languages are therefore not included.

**PROSPERO registration number** CRD42023390664 (https://www.crd.york.ac.uk/prospero/).

## INTRODUCTION

Worldwide, there is a constant increase in the extent and severity of mental illness,[1 2] exacerbated by the COVID-19 pandemic.[3 4] More than a quarter[5] of the world's population experiences a severe mental health problem at some point during their life. The burden of mental illness both on a socioeconomic level and on the quality of life is immense.[1 2 5] The WHO indicates that patients with chronic mental illness are 40–60% more likely to die 10–30 years earlier compared with people without a mental illness.[5]

People with mental illnesses have an increased risk of other somatic chronic illnesses.[6–8] Furthermore, for these people

maintaining a healthy lifestyle, financial stability and access to appropriate somatic care are often problematic.[9] Apart from (acute situations due to) their mental problems, exacerbation of somatic complaints in this group of people can result in recurring ambulance and emergency department care.[6 8 10–12] We use the term emergency care context (ECC) to reflect both ambulance care and Accident and Emergency department care.

To describe patients with mental health problems who seek emergency care, several definitions or terms are used in the scientific literature. These include patients with 'mental illness',[13] 'mental disorder'[14] or 'mental health crisis'.[15] Other authors focus on a specific subgroup of patients with mental health problems who need emergency care, such as patients with 'borderline personality disorders',[16] 'substance use disorders'[17] or 'suicide attempts'.[18] The problem with these descriptions is that uniformity is lacking. Moreover, some healthcare conditions (such as mental health disorders) are in many cases initially unknown and uncharted in the ECC. This adds to the reasons why ECC professionals typically focus on vital signs and behaviours in order to chart and treat acute physical healthcare problems. Therefore, the authors of this review protocol propose the concept 'mental dysregulation' for use in researching care for this vulnerable group in the ECC. The authors propose the following definition: 'In the ECC, patients with mental health problems who seek emergency care may experience disproportionate feelings like fear, anger, sadness or confusion, possibly with associated behaviours. When the aforementioned symptoms interfere with the patient's treatment or that of others, the patient is considered to have a mental dysregulation.' This concept and definition will be subject to a Delphi procedure including international experts on this topic in 2023–2024 for further harmonising concepts used in international research on this matter.

The care for patients with a mental dysregulation who seek emergency care can be complex and intensive. In addition to a physical disorder or illness, these patients may be confused or emotionally unstable,[8 9 19–23] which may hinder the identification and treatment of both physical and mental health symptoms. The provision of care for patients with a mental dysregulation can evoke a wide variety of feelings in ECC staff, ranging from worry or pity, to annoyance and frustration.[8 24–27] These feelings may originate from several challenges in the ECC for these patients. First, these challenges may be induced by the still existing and experienced segregation between somatic and mental treatment, long waiting times[28] and revisits of patients with a mental dysregulation which cause irritation and a decrease in confidence in the effectiveness of interventions. Second, healthcare professionals in the ECC do not always have the right knowledge of psychopathology and legislation[21 29] or the skills to use de-escalation techniques in a timely fashion,[22 30] which can lead to frustration.

These challenges and feelings may lead to stigma towards patients with a mental dysregulation who seek emergency care.[26–31] Several qualitative studies have revealed that patients with mental problems often feel stigmatised by healthcare professionals in the ECC.[15 31–33] A recent scoping review of factors that influence emergency department nurses care for patients with mental illness revealed that the emergency department environment greatly impacts the role of emergency department nurses, that their beliefs and perceptions lead to role conflicts, and the lack of knowledge and confidence impacts their view towards patients with mental illness. Specifically, some nurses reported they avoided patients, judged them as less important and attention seeking,[25] which implies stigma towards this group of patients.

In their landmark paper, Link and colleagues define stigma as existing when elements of labelling, stereotyping, separating ('us' from 'them'), status loss and discrimination co-occur in a power situation that allows these processes to unfold.[34] Thornicroft and colleagues use this definition and elaborate on four different types of stigma of which, for this review, interpersonal stigma (also referred to as 'public stigma') is important. Interpersonal stigma refers to the link between stereotypes, negative attitudes (prejudice) and negative behaviour (discrimination) towards people with mental health conditions. Three related components are part of interpersonal stigma: knowledge (ignorance or misinformation), attitudes (negative emotional reactions, such as prejudice) and behaviour (such as avoidance or rejection).[35]

Various interventions have been developed that impact attitude and behaviour and minimise stigma held by healthcare professionals. Thornicroft and colleagues report on 56 reviews, documenting changes in knowledge, attitudes and clinical skills as well as clinical confidence and self-efficacy as stigma-related outcomes. Several types of interventions are mentioned, such as e-interventions, use of simulations (eg, standardised role-plays with actors) and social contact, such as filmed or live recovery testimonials. Studies involving different methods of contact (eg, live or filmed) were more often associated with better outcomes for knowledge and attitudes when compared with educational interventions alone. Interventions including experts with lived experience in the design and evaluation of stigma interventions demonstrated better outcomes.[35] The interventions outlined from the reviews covered various populations (eg, mental health professionals, healthcare providers) and various mental health problems (eg, dementia care, suicide prevention, borderline personality disorder); only one publication was specific to emergency department healthcare staff and focused on their attitudes towards patients with substance-related presentations.[36] To our knowledge, a comprehensive review of the available scientific evidence in this specific ECC is lacking.

## AIM

To review the body of evidence to gain insight into interventions targeting healthcare professionals that minimise

stigma including beliefs, attitudes and behaviour towards patients with a mental dysregulation within the ECC.

## CONCEPTS

For this review, we define *patients with a mental dysregulation* in the ECC as follows: patients with mental problems who seek emergency care may experience disproportionate feelings like fear, anger, sadness or confusion, possibly with associated behaviours. When the aforementioned symptoms interfere with the patient's treatment or that of others, the patient is considered to have a mental dysregulation.

We applied the WHO definition of *intervention*, namely, 'an act performed for, with or on behalf of a person or population whose purpose is to assess, improve, maintain, promote, or modify health, functioning or health conditions'.[37]

*Stigma* is defined as the link between stereotypes, negative attitudes and discrimination against people with mental health conditions in society.[35]

*Interpersonal stigma* encompasses the association between stereotypes, adverse attitudes (prejudice) and detrimental actions (discrimination) directed at individuals dealing with mental health conditions. It comprises three interconnected elements: knowledge (comprising ignorance or misconceptions), attitudes (comprising unfavourable emotional responses, like prejudice) and behaviour (encompassing actions such as avoidance or rejection).[35]

The ECC in this review is defined as emergency care from the scene of acute incident of illness, followed by ambulance transport to and treatment in the emergency department.

*Healthcare professionals* include paramedics, emergency care nurses, nurse specialists, physicians and physician assistants who are directly involved in providing emergency care.

## RESEARCH QUESTIONS

1. Which interventions that address interpersonal stigma towards patients with a mental dysregulation have been designed, implemented and evaluated for healthcare professionals in the ECC?
2. From the healthcare professionals' perspective, which facilitators and barriers were encountered when implementing these interventions within the ECC?
3. From the service users' perspective, which facilitators and barriers were encountered when implementing these interventions within the ECC?

## METHODS AND ANALYSIS

We will perform an integrative review of empirical literature using methodological strategies proposed by Whittemore and Knafl,[38] using a seven-step framework outlined by Dhollande and colleagues[39] to identify current state-of-the-art knowledge on the topic and to identify current knowledge gaps in research. The seven steps applied to the proposed integrative review will involve the following: (1) write the review question; (2) determine the search strategy; (3) critically appraise the search results; (4) summarise the search results; (5) extract and reduce data; (6) analysis; and (7) conclusions and implications.[39] An integrative review allows for inclusion of diverse study methodologies, including qualitative, quantitative, experimental and mixed-methods studies, which makes it suitable for topics that have not been studied intensively. Results from an integrative review provide a complete and varied perspective of the studied topic.

To further enhance the quality of the review, the Preferred Reporting Items for Systematic Review and Meta-Analysis Protocols (PRISMA-P) recommendations[40] will be followed where applicable. The PRISMA-P checklist is attached as online supplemental appendix I. Software applications RefWorks-ProQuest, Rayyan and ATLAS.ti will be used to enhance the quality of the review, efficiency of the research work, and transparency of process and results.

### Patient and public involvement

As Thornicroft and colleagues highlight in their publication on stigma, it is important to involve people with lived experience of mental health conditions (PWLE) in both research and innovation projects.[35] For that reason, one or two PWLE will be included in the project team and coauthor the review. For the development of this review protocol, no patient or PWLE were involved.

### Data sources and search strategy

A systematic search will be applied, followed by identifying possible missed publications via the reference lists of included publications. The following scientific databases will be systematically searched for English and Dutch publications: CINAHL, PsycINFO and MEDLINE. Search terms will include 'Stigma', 'Interpersonal stigma', 'Public stigma', 'Interventions', 'Emergency healthcare', 'Emergency healthcare staff', 'mental health', 'emergency department', 'ambulance', 'paramedics' and related search terms, as used in the aforementioned databases. The full search strategy is attached as online supplemental appendix II. The results will be presented in the review. Additionally, the grey literature will be searched. This grey literature will be limited to national guidelines related to this topic (ECC) in English or Dutch. We will search via Google and via guideline organisations (eg, The National Institute for Health and Care Excellence, in the UK) and professional organisations (eg, the American Academy of Emergency Medicine).

### Study selection and sorting

Studies will be included using the following inclusion and exclusion criteria:

## Inclusion

- ► Original research report, both qualitative and quantitative studies.
- ► Randomised controlled trials, prospective and retrospective cohort studies, case–control studies, cross-sectional studies and, if available, literature reviews.
- ► Description of innovation projects related to the topic, depending on information on outcome variables (see the Data extraction, analysis and synthesis section below).
- ► Focused on intervention(s), targeting healthcare professionals, to reduce stigma on mental health problems held by ECC staff.
- ► Peer-reviewed publication.
- ► Published in English or Dutch.
- ► To obtain as much information as possible on relevant interventions, the search includes thesis and conference abstracts.

## Exclusion

- ► Abstract and/ or full text not available.

## Study selection

The results of the search will be assessed in the following steps: (1) review of title and abstract, and (2) full-text review of selected publications. Publications that clearly do not answer the research question or address the inclusion criteria will be excluded (ie, other topic, not an ECC, no stigma-related intervention, language). Both steps will be performed by two independent reviewers, using Rayyan, a web-based tool, designed for publication selection in scientific literature review.[41] When no consensus is reached on a particular publication, a third reviewer will make a final decision. A Preferred Reporting Items for Systematic Reviews and Meta-Analyses flow diagram will be used to provide a summary of the search and selection process.

## Quality appraisal

The quality of included studies will be assessed with the Mixed Methods Appraisal Tool,[42] developed and improved by Hong and colleagues.[43] Each included publication will be assessed by two independent reviewers. Quality appraisal is not used for study selection, but for means of interpreting the strength of findings and conclusions. As Thornicroft and colleagues[35] emphasise the importance of including persons with lived experience in the development of interventions, we will include this in reviewing and appraising interventions presented in the included publications, by answering this question: To what extent are people with lived experience involved in the design/ delivery/evaluation of the intervention and if involved, what were the results of this?

## Data extraction, analysis and synthesis

Data extraction will be performed using Atlas.ti, a software application, originally designed for qualitative research analyses purposes, but also used for transparent data extraction and analyses in literature review.[44] A predetermined set of labels will be set up within ATLAS. ti and will include (but is not limited to): definition of stigma, description of intervention, study population, study design, sample, site/setting, measurements, outcome variable(s), reported strengths of the study, reported limitations of the study.

The main outcome variables of interest are: (1) level of interpersonal stigma and (2) patient satisfaction. The secondary outcome variable of interest is: level of knowledge on psychiatry.

Depending on the instruments used in the original publications, the measures and instruments for results on the outcome variables will be listed, when appropriate, in table(s). In the discussion part of the review, the quality of the used instruments will be discussed.

Two independent reviewers will analyse the included publications, identifying (by adding the predefined labels to) citations in the publications, relevant for answering the review questions. Depending on the content of the results, we will, when appropriate, perform thematic analysis as described by Dhollande and colleagues.[39]

The selected citations will be further analysed and discussed by the authors, using ATLAS.ti tools, thereafter the extracted data will be used for answering the research questions.

## Ethics and dissemination

No ethical approval or safety considerations are required for this review. The proposed review will be submitted to a relevant peer-reviewed international journal. Results will be presented at relevant medical scientific conferences.

**Author affiliations**

[1]Institute of Nursing Studies, HU University of Applied Sciences Utrecht, Utrecht, The Netherlands
[2]University Medical Center Utrecht, Utrecht, The Netherlands
[3]University of Southampton, Southampton, UK
[4]Brain Centre, University Medical Centre Utrecht, Utrecht, The Netherlands
[5]School of Nursing and Midwifery, Griffith University, Gold Coast, Queensland, Australia
[6]Menzies Health Institute Queensland, Griffith University, Gold Coast, Queensland, Australia

**Contributors** GVdG, NG and WHWH developed the initial protocol: review questions, introduction and methods. NB, LS, FES, JC and MvV commented on the manuscript and rewrote sections of the manuscript. All authors documented their approval of the final version of the review protocol before submission. As senior researcher and project leader, WHWH is the guarantor of the review and review protocol.

**Funding** This review is part of the PONTAC Study (Psychisch ONTregeld in de Acute zorg/ Mental dysregulation in Acute healthcare). The PONTAC Study is co-financed by SIA-RAAK, part of the Netherlands Organization for Scientific Research (NWO), under award/grant number: RAAK.PUB08.029. Furthermore, this study was partly funded by a fellowship funding from the Netherlands Organisation for Health Research and Development (ZonMw), under award/grant number number 10040022110004, and was awarded to WHWH (Wietske Blom-Ham). The University of Applied Sciences, Utrecht; University Medical Centre, Utrecht; Diakonessenhuis, Utrecht; St Antonius Hospital, Utrecht; Meander Medical Center, Amersfoort; Regional Ambulance Service Utrecht, Altrecht; Mental Healthcare, Utrecht have contributed in-kind to the study. The latter seven institutions did not provide an award/grant number to this project.

**Competing interests** None declared.

**ORCID iD**
Geurt Van de Glind http://orcid.org/0000-0002-8647-6581

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
