## [Reviewer comments · BMJ Open]

ARTICLE DETAILS

TITLE (PROVISIONAL)	Review protocol for an integrative review on interventions to reduce interpersonal stigma towards patients with a mental dysregulation for ambulance and emergency department healthcare professionals
AUTHORS	Van de Glind, Geurt; Galenkamp, Niek; Bleijenberg, Nienke; Schoonhoven, Lisette; Scheepers, Floortje E; Crilly, Julia; van Veen, Mark; Ham, Wietske H.W.

VERSION 1 – REVIEW

REVIEWER	Rolfe, Ursula Bournemouth University, HSS
REVIEW RETURNED	10-May-2023

GENERAL COMMENTS	Feedback on Manuscript: Integrative Review on Interventions to reduce Interpersonal Stigma towards patients with a mental dysregulation for ambulance and emergency department healthcare professionals Comments: Page 2 Line 7: why is underlying in brackets? Lines 7-8: Consider rewriting this sentence with a removal of brackets and hyphens to improve flow and syntax. Line 10: you begin with mental illness and then use the term mental dysregulation. Perhaps consider consistency or further context. Line 48: 'more than a quarter or even half'...this is confusing, there is a big difference between quarter and half. Page 3 Line 3: same as above, why is underlying in brackets? Line 45: you refer to acute healthcare workers but also make reference to emergency care, I assume you mean to differentiate these roles, if not please chose one or the other for consistency. Line 55: remove brackets from timely
--

	Page 4 Line 15 and 18: Is this referencing correct? Page 5: Line 37: what about occupational therapists and physiotherapists and healthcare assistants? All work in ED. Line 45: What is Interpersonal Stigma in caps? Page 6 Check referencing here, seems to be a mixture...? Line 15: check punctuation Page 7 Line 19: why are Healthcare Professionals in caps? Line 41: Check punctuation why is the hyphen here? Quality appraisal section: be consistent with referencing style Page 8 Line 23: check caps in this sentence again
--	---

VERSION 1 – AUTHOR RESPONSE

Reviewer 1 Comments with added author's responses:

Feedback on Manuscript: Integrative Review on Interventions to reduce Interpersonal Stigma towards patients with a mental dysregulation for ambulance and emergency department healthcare professionals

Comments:

Page 2

Line 7: why is underlying in brackets?

AR: We can see the confusion these brackets might cause. As a solution we decided to remove the word 'underlying'.

Lines 7-8: Consider rewriting this sentence with a removal of brackets and hyphens to improve flow and syntax.

AR: Based on the previous comment and our response, we decided to remove the word 'underlying'. We changed the sentence, based on the reviewers' comment as follows: "An exacerbation of somatic complaints in this group of people can result in recurring ambulance and emergency department care".

Line 10: you begin with mental illness and then use the term mental dysregulation. Perhaps consider consistency or further context.

AR: We thank the reviewer for this comment which reflects our own discussions on these related terms. We have added clarity in the abstract to help explain this, with the sentence now reading: "The care of patients with a mental dysregulation (i.e., experiencing a mental health problem and disproportionate feelings like fear, anger, sadness, or confusion, possibly with associated behaviours) can be complex in the emergency care context, possibly evoking a wide variety of feelings, ranging from worry or pity to annoyance and frustration in emergency care staff members."
To overcome word-count limits for the abstract we adjusted several sentences in the abstract, without changing content or meaning of these sentences.

Line 48: 'more than a quarter or even half'...this is confusing, there is a big difference between quarter and half.

AR: We agree with the reviewer. We have adjusted the sentence to stick to the peer-reviewed reference of over 25% (Kessler RC, Aguilar-Gaxiola S, Alonso J, Chatterji S, Lee S, Ormel J, et al. The global burden of mental disorders: an update from the WHO World Mental Health (WMH) surveys; PMC3039289. *Epidemiol.Psichiatr.Soc.* 2009;18(1):23-33.) and removed 'or even half'.

Page 3

Line 3: same as above, why is underlying in brackets?

AR: : Based on the previous comment and our response, we decided to remove the word 'underlying'. We changed the sentence, based on the reviewers' comment as follows: "An exacerbation of somatic complaints in this group of people can result in recurring ambulance and emergency department care".

Line 45: you refer to acute healthcare workers but also make reference to emergency care, I assume you mean to differentiate these roles, if not please chose one or the other for consistency.

AR: We thank the reviewer for this comment and agree that we use several terms inconsistently throughout our manuscript. To overcome this, we have added the following sentence to the first paragraph of the Introduction: "We use the term *Emergency Care Context (ECC)* to reflect both Ambulance care and Accident and Emergency department care." And we have changed other used concepts describing the **context** (ambulance and Accident and Emergency department care) accordingly, using the abbreviation 'ECC' throughout the manuscript. When talking about patients seeking 'emergency **care**', we decided to stick to these wordings.

Line 55: remove brackets from timely

AR: We have done so.

Page 4

Line 15 and 18: Is this referencing correct?

AR: We checked the related references and the concepts are in the titles of the related papers. concepts. In addition, all references used here are related to the emergency care context (i.e., ambulance or Accident and Emergency department). So we do think the referencing is correct.

Page 5:

Line 37: what about occupational therapists and physiotherapists and healthcare assistants? All work in ED.

AR: These professions are indeed not included in this review. We decided to limit our search to the professions mentioned in the review protocol.

Line 45: What is Interpersonal Stigma in caps?

AR: Whilst the concept of stigma is likely familiar to the target audience of our review, the concept of interpersonal stigma is (in our view) not familiar. We have therefore added a definition of interpersonal stigma to the CONCEPTS section: "Interpersonal stigma encompasses the association between stereotypes, adverse attitudes (prejudice), and detrimental actions (discrimination) directed at individuals dealing with mental health conditions. It comprises three interconnected elements: knowledge (comprising ignorance or misconceptions), attitudes (comprising unfavorable emotional responses, like prejudice), and behavior (encompassing actions such as avoidance or rejection) (37)."

We have also revised the manuscript changing our use of capitals, revising 'Interpersonal Stigma' to 'interpersonal stigma'.

Page 6

Check referencing here, seems to be a mixture...?

AR: We thank the reviewer for this comment and apologize for our lack of punctuality. We have changed the referencing appropriately on page 6, and checked the entire manuscript for similar inappropriate referencing.

Line 15: check punctuation

AR: Thanks again for this comment. We have done so.

Page 7

Line 19: why are Healthcare Professionals in caps?

AR: We do not recall a particular reason. We therefore agree with this comment and removed the capital letters and changed these to regular letters.

Line 41: Check punctuation why is the hyphen here?

AR: We do not recall a particular reason. We therefore agree with this comment and removed the hyphen.

Quality appraisal section: be consistent with referencing style

AR: as mentioned in the previous comment on referencing style, we have adjusted this accordingly throughout the manuscript.

Page 8

Line 23: check caps in this sentence again

AR: We have checked the entire manuscript for capitals used in the term 'interpersonal stigma', and removed the capitals and changed these in regular letters, when appropriate. In addition we removed the capital in the concept 'patient satisfaction' in this sentence.

Additional author's response: adjustment of the search strategy.

Along with the development of the review protocol we submitted to BMJ-open early this year, we developed the search strategy for our review. It was added as Appendix II to the submitted manuscript.

We started our search and inclusion process upon submission, as indicated in the 'Planning and time-frame'-section in the review protocol. In the months we did so, Artificial Intelligence was in the media a lot. And one of our review-team members heard about 'Evidence Hunt', an Artificial Intelligence application added to the Medline database. He decided to ask Evidence Hunt a question related to

our research question. Much to our surprise an answer was given with several references. The most of these were present in the search we performed based on our search strategy. But there were also a few references that were not in our search. We decided to discuss this finding with our team and our librarian. We carefully looked at our search strategy, and at the key words used in these additional interesting publications (found via Evidence Hunt). This resulted in a slightly adjusted search strategy:

- We have removed the - AND (Patients[Mesh] OR Patient*[tiab] OR Client*[tiab]) – section.
- We have added the term “First Responder”
- We have added terms related to “Suicide” and “Self-Injurious Behavior”

As we broadened our search, we made sure that the results of the previous search was part of the second search we performed 13th July 2023. We will therefore use the same description of the search strategy as mentioned in the review protocol as it was submitted 8th February 2023. But we will use the updated search strategy as Appendix II.